# Multimodal, label-free fluorescence and Raman imaging of amyloid deposits in snap-frozen Alzheimer's disease human brain tissue

Benjamin Lochocki [1✉], Baayla D. C. Boon [2], Sander R. Verheul[1], Liron Zada[1], Jeroen J. M. Hoozemans[2], Freek Ariese[1] & Johannes F. de Boer[1]

Alzheimer's disease (AD) neuropathology is characterized by hyperphosphorylated tau containing neurofibrillary tangles and amyloid-beta (Aβ) plaques. Normally these hallmarks are studied by (immuno-) histological techniques requiring chemical pretreatment and indirect labelling. Label-free imaging enables one to visualize normal tissue and pathology in its native form. Therefore, these techniques could contribute to a better understanding of the disease. Here, we present a comprehensive study of high-resolution fluorescence imaging (before and after staining) and spectroscopic modalities (Raman mapping under pre-resonance conditions and stimulated Raman scattering (SRS)) of amyloid deposits in snap-frozen AD human brain tissue. We performed fluorescence and spectroscopic imaging and subsequent thioflavin-S staining of the same tissue slices to provide direct confirmation of plaque location and correlation of spectroscopic biomarkers with plaque morphology; differences were observed between cored and fibrillar plaques. The SRS results showed a protein peak shift towards the β-sheet structure in cored amyloid deposits. In the Raman maps recorded with 532 nm excitation we identified the presence of carotenoids as a unique marker to differentiate between a cored amyloid plaque area versus a non-plaque area without prior knowledge of their location. The observed presence of carotenoids suggests a distinct neuroinflammatory response to misfolded protein accumulations.

[1] Department of Physics and Astronomy, LaserLaB Amsterdam, VU Amsterdam, Amsterdam, The Netherlands. [2] Department of Pathology, Amsterdam Neuroscience, Amsterdam UMC—location VUmc, Amsterdam, The Netherlands. ✉email: ben.lochocki@vu.nl

Dementia affects memory, cognitive abilities, and behavior of mainly elderly people. Alzheimer's disease (AD) is the most common type and contributes to around 65% of the dementia cases[1]. Together with hyperphosphorylated tau deposits, the accumulation of amyloid-beta (Aβ), commonly referred to as amyloid plaque, is one of the main pathological hallmarks of AD[2–4]. Within the plaque, misfolded protein chains, cleaved off from the amyloid precursor protein, aggregate in an insoluble, anti-parallel β-sheet structure which subsequently leads to different pathological amyloid-plaque types[5]. Attempts at in vivo diagnosis include the targeting of Aβ in the brain[6–8], in cerebrospinal fluid (CSF)[9–11], and recently also in blood or serum[11–15]. However, these techniques always rely on the indirect binding of Aβ by an antibody or labeling compound and thus provide no specific information on the intrinsic properties of the pathological substrate. Although at least partially invasive, these methodologies cannot provide a definitive diagnosis of AD. At the moment, conclusive assessment of AD can only be obtained post-mortem by (immuno-) histochemical staining on brain tissue[16,17]. Staining protocols are often labor-intensive and time-consuming, and there may be risk of dehydration, heat, and deformation. A label-free characterization of the molecular composition of fresh tissue could provide a fast assessment of pathology and potentially an option for in vivo AD diagnosis in, e.g., the retina[17,18]. Such non-invasive or minimally invasive techniques include among others Fourier-transform infrared spectroscopy[19], coherent anti-Stokes Raman spectroscopy[20], and surface-enhanced Raman spectroscopy[21]. These techniques could generate new possibilities to detect and follow pathology while giving new insights in the composition of the pathological substrate. While in most of the available AD-related literature and research, transgenic AD mice were used, Hodge et al.[22] stressed the importance of studying human tissue as the differences from cell types in mouse tissue are extensive. Recently, we investigated whether Raman spectroscopy ($\lambda_{exc} = 785$ nm) could be used for the detection of Aβ plaques in fixed human brain tissue. Despite the potential of vibrational spectroscopic techniques for a label-free and non-invasive characterization of biomolecular composition, a unique Aβ-associated spectrum could not be determined[23]. In the present study, we report on data obtained from post-mortem, snap-frozen human AD tissue. We used auto-fluorescence microscopy, followed by spontaneous Raman spectroscopy using a 532 nm excitation source to examine the unstained plaque tissue sections. Afterward, we recorded stimulated Raman scattering (SRS)[24,25] images of the same tissue areas. To the best of our knowledge, it is the first time that various imaging and spectroscopic modalities were used consecutively on the same native AD tissue section. Moreover, each examined tissue section was eventually stained with thioflavin-S to confirm plaque pathology and the exact locations within the tissue.

## Results and discussion
The results are presented in the following order: In section Fluorescence microscopy, we demonstrate that amyloid plaques can be identified based on their (auto-) fluorescence properties in native-unstained tissue confirmed by subsequent thioflavin-S staining. In section Spontaneous Raman spectroscopy we show that cored amyloid accumulations can be identified by conventional Raman spectroscopy using 532 nm excitation. We identified unique spectral hallmarks that enabled us to detect plaques within tissue without prior knowledge of their location. These spectral features are further characterized in section Resonance enhancement. Thirdly, we present SRS measurements in section SRS microscopy, looking specifically for a protein peak shift when measuring within an amyloid plaque compared to blank background tissue.

It is worth noting that all samples were subject to the same conditions: the Raman settings, the hardware used, the laser power reaching the samples, the staining protocol, and data processing were identical to reduce possible experimental variability. The same is true for the SRS measurements, where objective, laser power, and pixel dwell time were kept constant. The experimental work flow is illustrated in Fig. 1. Table ST1, presented in the Supplementary Information, gives an overview, which imaging modality was applied to which section.

**Fluorescence microscopy**. In Fig. 2, we show a column-wise overview of three measured tissue sections, characterized in Table 1. The remaining cases are shown in Fig. S3. Figure 2a shows the auto-fluorescence images of the freshly cut tissue, acquired with the fluorescence microscope using a ×20 and ×40 objective for the first and second row, respectively. The bright, yellow/orange and uniformly distributed spots, which can be seen in all auto-fluorescence tissue samples (some spots are indicated by blue arrows in Figs. 2a and 3) are lipofuscin granular deposits[23,26]. Lipofuscin, an age-related metabolic waste product, is highly unlikely an indicator or hallmark of AD since it is also found in control cases with a similar distribution[23,27] (see also controls in Fig. S4). Lipofuscin auto-fluorescence properties are well known from brain tissue[26,28–33] and retinal imaging[34,35] and are not further investigated here.

In the areas of interest (yellow and red dashed boxes), the auto-fluorescence of amyloid accumulations appears greenish (seen best in the second row of Fig. 2a), either in a compact and dense shape (#1a, #1b) or as a cloudy fibrillar haze (#2). It should be mentioned that these green spots were more easily spotted by eye under the microscope than in the recorded RGB images. Some areas show a faint dark greenish patchy background without substantial features. In order to provide further evidence, the enlarged images of the red dashed boxed areas are shown in the Supplementary Information in Fig. S2.

In Fig. 2b the fluorescence images of the same tissue sections are shown after thioflavin-S staining to highlight amyloid deposits. The positively stained thioflavin-S areas (bright yellow areas) match perfectly the greenish auto-fluorescent areas, while the rest of the tissue is observed as a uniform dark background. The co-localization of the thioflavin-S staining and the green auto-fluorescence tissue areas indicates that these are amyloid plaques. The auto-fluorescence signal of the lipofuscin that was bright in the images of Fig. 2a now remains hidden in the background due to the high thioflavin-S fluorescence signal and the reduction in auto-fluorescence of lipofuscin over time (but is still sometimes seen as weak, light orange fluorescence). The additional cases are shown in Fig. S3.

Figure 3 is an enlarged, side-by-side comparison of distinct plaque locations (yellow dashed boxes in Fig. 2 for the cases #1a, #1b, and #2), comparing the auto-fluorescence images with the subsequently acquired fluorescence images after thioflavin-S staining. Figure 3 confirms the perfect match between greenish auto-fluorescence area and the thioflavin-S positive staining of the tissue. No green emission was observed in the tissue samples from control cases (n = 5, see Fig. S4).

The greenish appearance observed in the present study agrees with a recent publication on AD mice tissue where various excitation sources in the range of 460–490 nm were used and plaque locations emitted in green (510 and 530–550 nm)[36]. Others reported similar results when illuminating with a 475 ± 10 nm source. Due to their additional NIR measurements they concluded amyloid fibers as source of the luminescence, while stating that "the underlying basic phenomena remain mostly unknown"[30]. However, the authors could not exactly specify

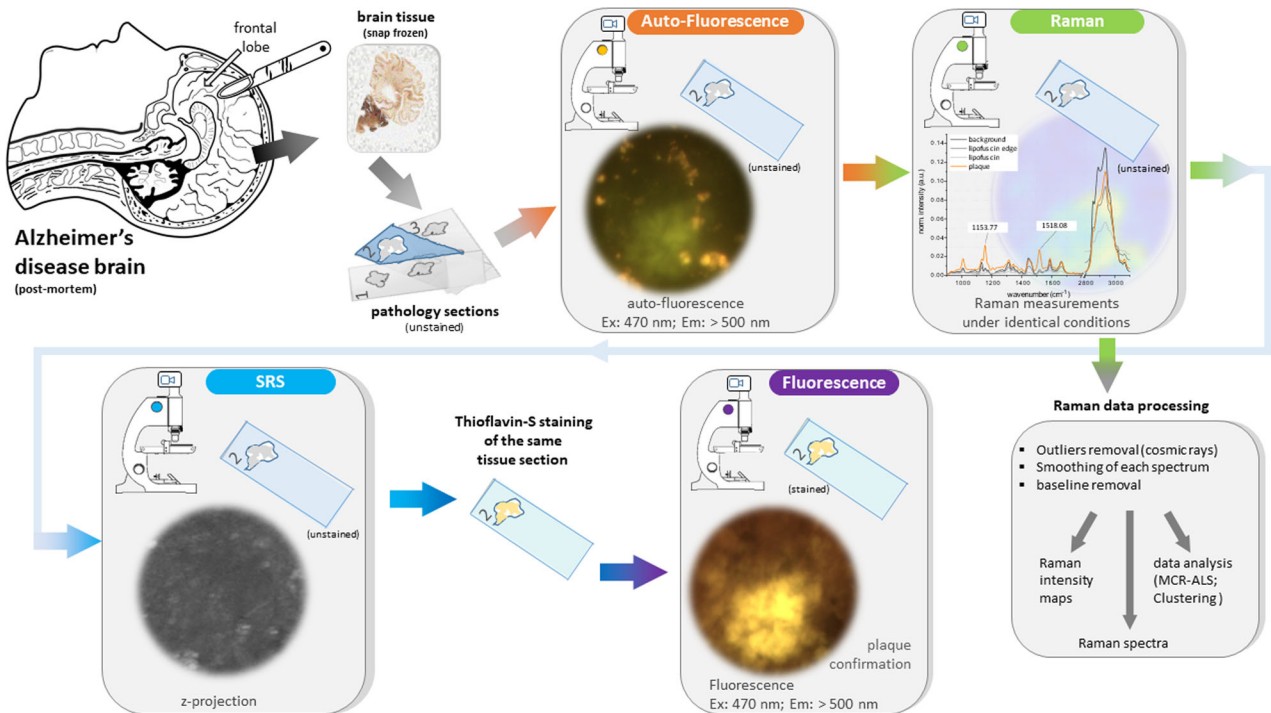

**Fig. 1 Experimental work flow.** Brain autopsy is performed within 9 h post-mortem and the tissue is immediately snap-frozen to −80 °C. Next, the tissue is cut at 20 μm and mounted on a CaF$_2$ microscope slides. Afterwards, the section is imaged under the fluorescence microscope. Subsequently, Raman maps are recorded. Hereafter, the tissue is measured across the amide-I protein peak using the SRS microscope. Following SRS, the tissue section is stained with thioflavin-S and imaged under the fluorescence microscope for plaque confirmation. Finally, the data are processed and analyzed.

the peak emission wavelength since the emission was recorded with a rather broad bandpass filter, ±50 nm, centered at 530 nm. Thal et al.[37] measured auto-fluorescence when illuminating human AD brain tissue with UV (330–385 nm) and deep blue (400–410 nm) and recorded emission images above 420 and 455 nm, respectively; therefore, the plaques appeared in (dark) blue. However, no emission curve was presented by the authors.

To determine the spectral response of amyloid deposits we measured the green emission with a confocal microscope in spectral detection mode, using an excitation source of 488 nm, close to the 470 nm LED of the full-field fluorescence microscope. Amyloid deposits in adjacent sections of cases #3 and #4 were investigated, resulting in a total of 3 cored and 2 fibrillar amyloid deposits. In Fig. 4 the results for one of the cored and one of the fibrillar amyloid deposit measurements are shown. The emission of the core plaque peaks around 540 nm and for the fibrillar plaque around 549 nm (mean values). Lipofuscin peaks in both cases around 566 nm. Similar emission curves of the three additionally measured plaques are shown in the Supplementary Information in Fig. S5.

It is important to mention that we also observed a few greenish auto-fluorescence areas of different appearance which did not stain positive for amyloid deposit with thioflavin-S. These looked similar in terms of density and color but had a spherical shape and were smaller with a diameter of 10–15 μm. In Fig. S6 (Supplementary Information) we show two of those spots, comparing the auto-fluorescence before and the fluorescence image after thioflavin-S staining. As can be noted, the green bubble-like structures are not stained by thioflavin-S and are therefore not positive for amyloid. We hypothesize that these intracellular bodies are corpora amylacea. In agreement with the literature, the observed cellular bodies match size, luminescence, and the lack of Aβ[38–40]. Furthermore, corpora amylacea are known for their

abundance in neurodegenerative diseases[39] and were recently proposed as "containers to remove waste products"[41].

**Spontaneous Raman spectroscopy.** After recording the Raman maps, the spectral data were analyzed and visualized. In columns 2–6 in Fig. 5, the processed images are compared to the acquired (auto-) fluorescence images of column 1. The Raman mapping images were taken in 1-μm step size and cover an area of 61 × 61 μm for plaques #1a, #1b, #3a, and #3b, 201 × 201 μm for plaque #2, 81 × 81 μm for plaques #3b and 5, and 101 × 101 μm for #4. The overview for the control cases can be found in the Supplementary Information, Fig. S7. The total intensity images (meaning the integrated spectra per pixel) are shown in column 2 of Fig. 5, emphasizing areas with underlying fluorescent lipofuscin granules (orange-red spots in the images in the first column) while the rest of the tissue shows an almost uniform background. Columns 3 and 4 show the Raman peak intensity images (after data pre-processing but before MCR-ALS and clustering) of the protein and lipid bands. In addition, the ratio maps[42] of plaques #1a and #1b are shown in the Supplementary Information (Fig. S8). The protein and lipid mapping images highlight, similar to the total intensity images but in an inverse manner, the granular lipofuscin accumulations but they do not provide any visual evidence of potential increased β-sheet deposits. For the areas that were at a later stage confirmed with thioflavin-S staining to contain amyloid plaque (see below) we then looked for any substantial changes in the Raman spectra at protein wavenumbers. This was triggered by other studies that provided evidence of a distinct protein peak shift towards the expected β-sheet peak when using either SRS on mice tissue[24] or conventional Raman scattering on human AD brain tissue[43] or CSF and blood samples[44]. In the Supplementary Information, Fig. S9, we show the Amide-I band (1640–1680 cm$^{-1}$) for all AD cases, where we compare the Raman spectrum of plaque locations to the averaged

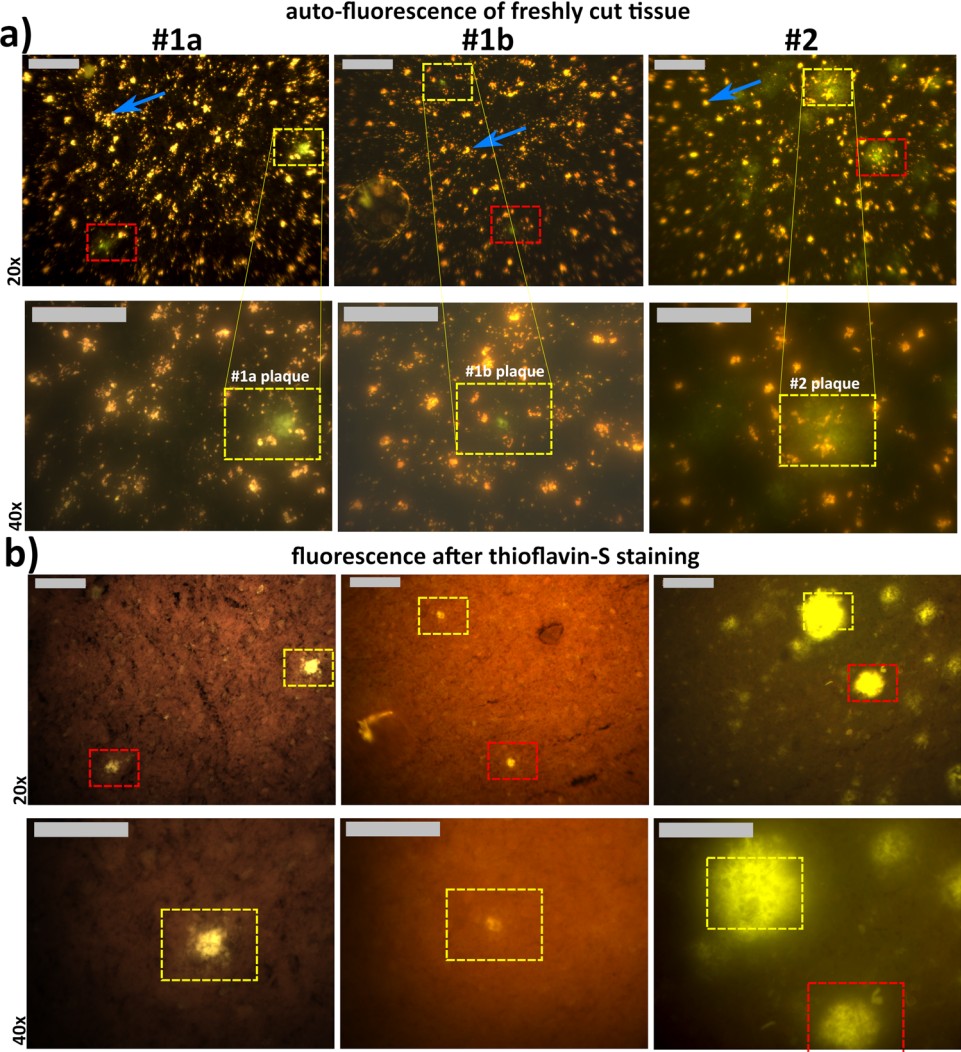

**Fig. 2 Fluorescence images of AD cases 1 and 2, before and after staining, at ×20 and ×40 magnification. a** Auto-fluorescence images of freshly cut tissue. Greenish dense cores (#1a and #1b) and greenish patches (#2) are clearly visible. The uniformly distributed yellow spots (blue arrows, just one is identified in each image) are lipofuscin deposits[23]. **b** Fluorescence images of the same tissue sections and areas stained with thioflavin-S. Here, the yellow bright spot areas are the positively stained amyloid accumulations. All images were taken with the full-field fluorescence microscope and an illumination source of 470 nm. Yellow and red dashed boxes mark the measured plaque areas. # refers to AD donor number and plaque in Table 1. Scale bars: 100 µm. The ×40 image pairs of the red dashed boxes are shown in the Supplementary Information, Fig. S2. Enlarged images of each plaques are shown in Fig. 3. The image pairs of cases 3–5 are shown in Fig. S3.

non-plaque spectrum, based on the MCR-ALS cluster spectra. The detected β-shifts were not as prominent as expected and were not found in all AD cases. Although the overall study size was limited, our results suggest that conventional Raman mapping at protein or lipid wavenumbers or peak ratio imaging, as commonly used for tissue analysis[45–47], does not appear to be such a robust method for the detection of amyloid plaques in brain tissue.

On the other hand, strong Raman peaks at 1518 and 1154 cm$^{-1}$, indicative of carotenoid compounds, were detected specifically at the cored plaque locations, as shown in the peak intensity images in columns 5 and 6 of Fig. 5. These two unique Raman peaks enable the spectral differentiation of the cored plaque areas from the surrounding tissue. This is further illustrated in Fig. 6, where we plot the full spectra (orange lines) associated with each plaque. The spectral origin is discussed in more detail in section Resonance enhancement. In none of the plaques of AD donor #2 and #4 the carotenoid-specific spectrum was observed. The measured plaques in those two cases were of the fibrillar plaque

type. Only one fibrillar plaque (#3b) showed clear carotenoid peaks within the fibrillar amyloid deposit. Interestingly, in this AD donor the majority of plaque types was of the dense cored type whereas in case #2 and #4 the majority of plaques were of the fibrillar type. This suggests that the carotenoid signal differs in plaque types and might especially be associated with dense cored plaques.

The last column in Fig. 5 shows the fluorescence images of the same tissue slice and area after thioflavin-S staining. Amyloid deposits are stained positively in yellow, matching not only the green auto-fluorescence areas in the unstained images but also the Raman peak intensity images (1154 cm$^{-1}$ and 1518 cm$^{-1}$) at carotenoid wavenumbers and the cluster images column 7) and therefore confirm the actual presence of plaques in the measured tissue areas.

In Fig. 6 we compare the obtained spectra after clustering of the MCR-ALS data, depicting the plaque-associated spectra in orange. In all cored plaques (#1a, #1b, #3a, #3c, #5) and in plaque #3b we have an identical spectral distribution with the peaks

**Table 1 Demographics of AD brain donors and imaged plaques.**

| AD donor/control | Sex | Age[a] | Braak[88] | Amyloid[88] | Region | PMD[b] | Location (plaque reference number) | Plaque type |
|---|---|---|---|---|---|---|---|---|
| #1 | Female | 90 | 5 | C | Middle frontal gyrus | 4 | a | Cored amyloid deposit |
| | | | | | | | b | Cored amyloid deposit |
| #2 | Male | 72 | 5 | C | Middle frontal gyrus | 6 | | Fibrillar amyloid deposit |
| #3 | Male | 84 | 4 | B | Middle frontal gyrus | 5:53 | a | Cored amyloid deposit |
| | | | | | | | b | Fibrillar amyloid deposit |
| | | | | | | | c | Cored amyloid deposit |
| #4 | Male | 65 | 6 | C | Superior frontal gyrus | 8:50 | a | Fibrillar amyloid deposit |
| | | | | | | | b | Fibrillar amyloid deposit |
| | | | | | | | c | Fibrillar amyloid deposit |
| #5 | Female | 84 | 6 | C | Middle frontal gyrus | 6:30 | | Cored amyloid deposit |
| c#1 | Female | 62 | 1 | B | Middle frontal gyrus | 7:55 | a | *Control* |
| | | | | | | | b | |
| | | | | | | | c | |
| c#2 | Male | 64 | 1 | A | Middle frontal gyrus | 8:25 | a | *Control* |
| | | | | | | | b | |

Measuring a total of 10 plaques in five different patients. The last two rows are the control cases.
[a]Age in years.
[b]PMD post-mortem delay in hours.

**Fig. 3 Side-by-side comparison of plaque locations (as indicated by the yellow dashed boxes in Fig. 2 for cases #1a, #1b, and 2) of auto-fluorescence images and the same location after staining with thioflavin-S.** The blue arrows point to lipofuscin deposits, which are not visible in the thioflavin-S stained images since the fluorescence of thioflavin-S is stronger. Scale bars: 40 μm.

around 1154 and 1518 cm$^{-1}$ (and also the elevated peak around 1008 cm$^{-1}$ which partially overlaps with the phenylalanine peak at 1003 cm$^{-1}$). These Raman peaks are well-known to be associated with carotenoids[48,49], which can be found in the human body, including in the brain[50] and also in the retinal macular pigment[51,52]. For further discussion and analysis of carotenoids in amyloid plaques, please see section Resonance enhancement. The spectral data of the background (dark and medium dark gray) differ only slightly in intensity with no peak variations observed. The background exhibits the main common tissue peaks at 1660 cm$^{-1}$ (protein), 1445 cm$^{-1}$ (lipids), and others. The lipofuscin spectrum (light gray) is the lowest in overall intensity but shows the same vibrational modes for proteins and lipids across the spectrum. Its apparent lower intensity can be explained as a consequence of the background removal algorithm. The raw spectral data of lipofuscin areas contain the highest fluorescence background and relatively little Raman scattering on top of it[23]. The background removal

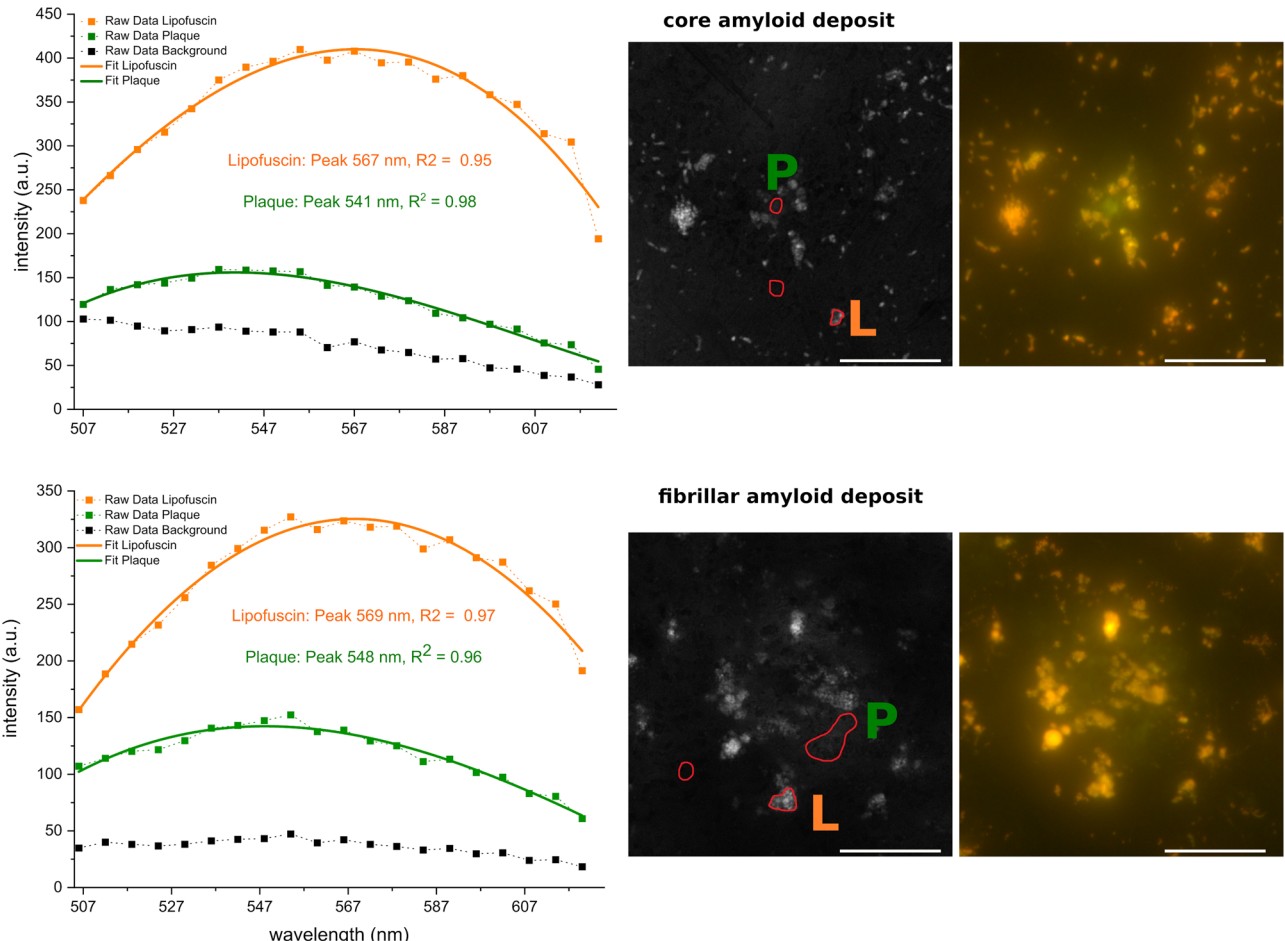

**Fig. 4 Fluorescence emission spectra of plaque areas excited with a 488 nm source.** Top row: cored amyloid deposit. Bottom: fibrillary amyloid deposit. Symbols connected with a dashed line are the measured data. The solid lines are the corresponding third-order polynomial fits; peak position and $R^2$ values are indicated. The right half shows the corresponding z-projection of the emission images and the locations where the data points for plaque (P), lipofuscin (L), and background were taken, next to it the corresponding full-field auto-fluorescence images. Corresponding thioflavin-S images are not shown. Scale bar: 40 µm.

algorithm subtracts most of the underlying fluorescence, resulting in a low-intensity Raman spectrum for the lipofuscin spots.

When analyzing the spectral clustering of fibrillar plaques #2 and #4 (a–c), the above-mentioned carotenoid peaks were not detected. The red area in the cluster image of #2 and #4c shows the points which largely cover the locations where a plaque was confirmed by thioflavin-S staining, but the associated spectra do not show any carotenoid peaks. All four-cluster spectra show roughly the same spectral distribution with very similar peaks and only minor intensity variations. The peaks at 1659 and 1439 cm$^{-1}$ are most likely slightly shifted Amide-I (protein) and lipid bands. Together with the 1005 cm$^{-1}$ phenylalanine peak, they represent the general Raman peaks of human brain tissue. Even though we do not observe carotenoid peaks in most of the fibrillar amyloid plaques, there are other spectral peaks, which are also present but to a lesser extent in the spectra of other plaques. Common peaks are at 760 cm$^{-1}$, associated with DNA or protein; at 1130 cm$^{-1}$, associated with C–N stretching of proteins and overlapping with the phosphatidylcholine lipid band; at 1311 cm$^{-1}$, the CH$_2$ twisting mode of lipid combined with the Amide-III mode, and at 1586 cm$^{-1}$, most likely associated with retinoid compounds[53,54]. The four-cluster spectra of the control cases were obtained in a similar manner but show no sign of carotenoid peaks. The cluster images and corresponding spectra of the control cases can be found in the Supplementary Information (Fig. S10).

**Resonance enhancement**. In the Supplementary Information, Fig. S12, we show a spectral Raman clustering comparison between the 532 and 785 nm excitation source of the identical tissue area (plaque #1b), demonstrating the absence of detectable carotenoid peaks when using the NIR source. Similar observations were reported before, when Raman spectroscopy with different excitation sources was used to study colon cancer tissue[55]. In our recent work[23], we were not able to reveal unique spectral differences between plaque and non-plaque areas within formalin-fixed AD tissue. Although in principle the fixation of the tissue might be responsible for the unsuccessful distinction of amyloid plaques, the more likely reason is the use of a different Raman excitation source of $\lambda = 785$ nm. In order to assess the effect of the Raman excitation wavelength, we recorded the absorption spectra of two carotenoid compounds β-carotene and lutein (Sigma-Aldrich) dissolved in hexane. We observed that the compounds have similar UV–VIS spectra, with two substantial peaks at around 450 (443) and 478 (472) nm, respectively (see Fig. S13 in the Supplementary Information) in agreement with literature[51,56,57]. Therefore, using a 532 nm Raman excitation source is expected to enhance the carotenoid Raman signals relative to those of the other tissue components through the so-called pre-resonance effect[58]. The green 532 nm laser lies at the edge of the absorption flank of carotenoids, whereas there will be practically no resonance enhancement with a NIR source. We

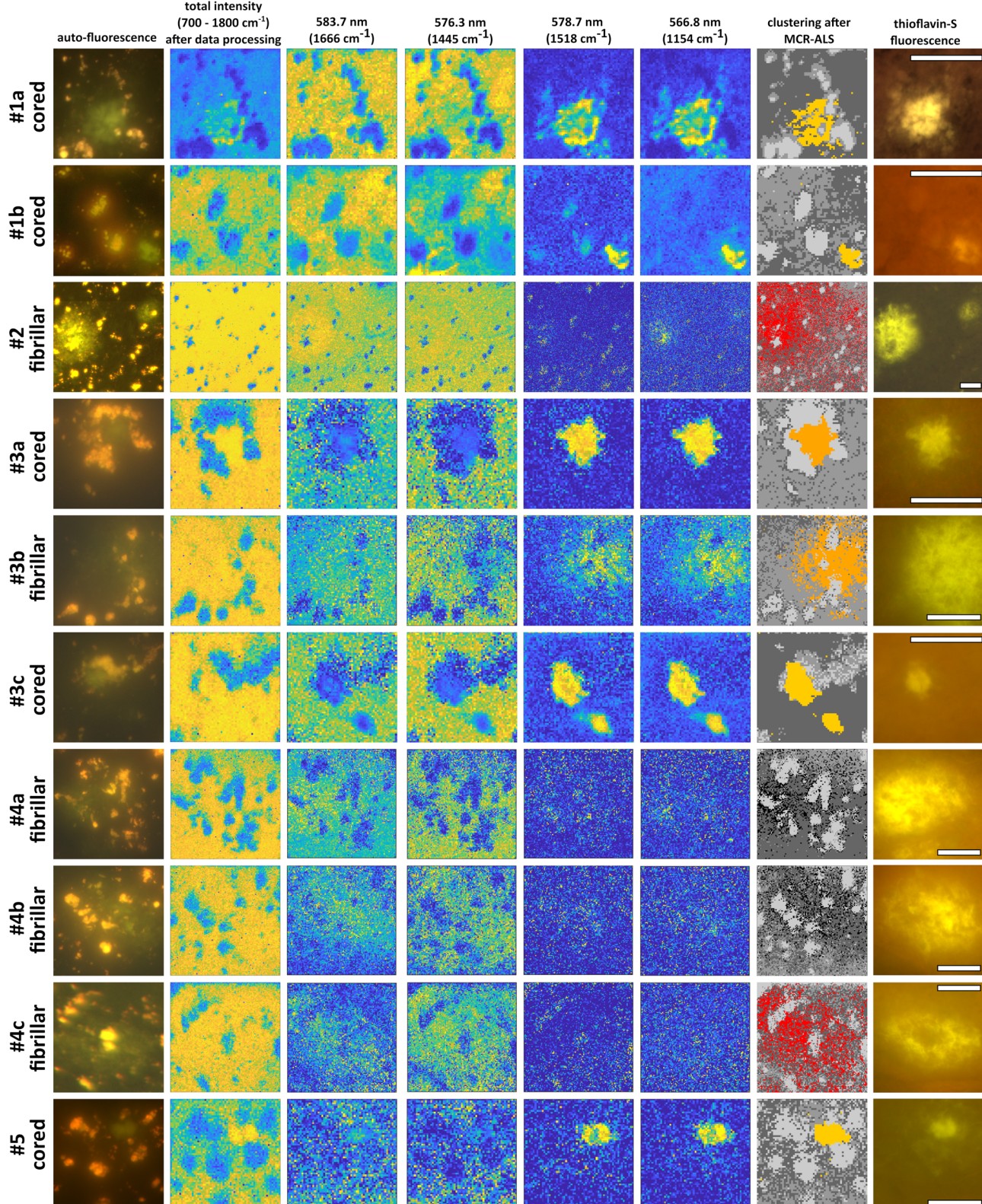

**Fig. 5 Overview of fluorescence and Raman images.** Each row represents one tissue section. First column: Auto-fluorescence images of the presumed plaque (green) areas. Second column: The total intensity image of the spectral Raman data after data pre-processing. Third and fourth column: Raman peak intensity images of the protein (1666 cm$^{-1}$) and lipid (1445 cm$^{-1}$) bands. Fifth and sixth column: Raman peak intensity images at two prominent carotenoid wavenumbers (1518 and 1154 cm$^{-1}$). Seventh column: four-cluster images of the Raman data after MCR-ALS computing, highlighting the plaque locations in orange (#1) and red (#2 and #4c). Last column (eighth): Fluorescence image of the thioflavin-S stained tissue, confirming anticipated plaque locations by yellow fluorescence. Note for cored plaques #1a, #1b, #3a, #3c, and #5 how well the carotenoid peak images and the cluster analysis match the stained plaque depositions. Scale bars: 40 μm; color coding for columns 2 to 6: blue (low) to yellow (high).

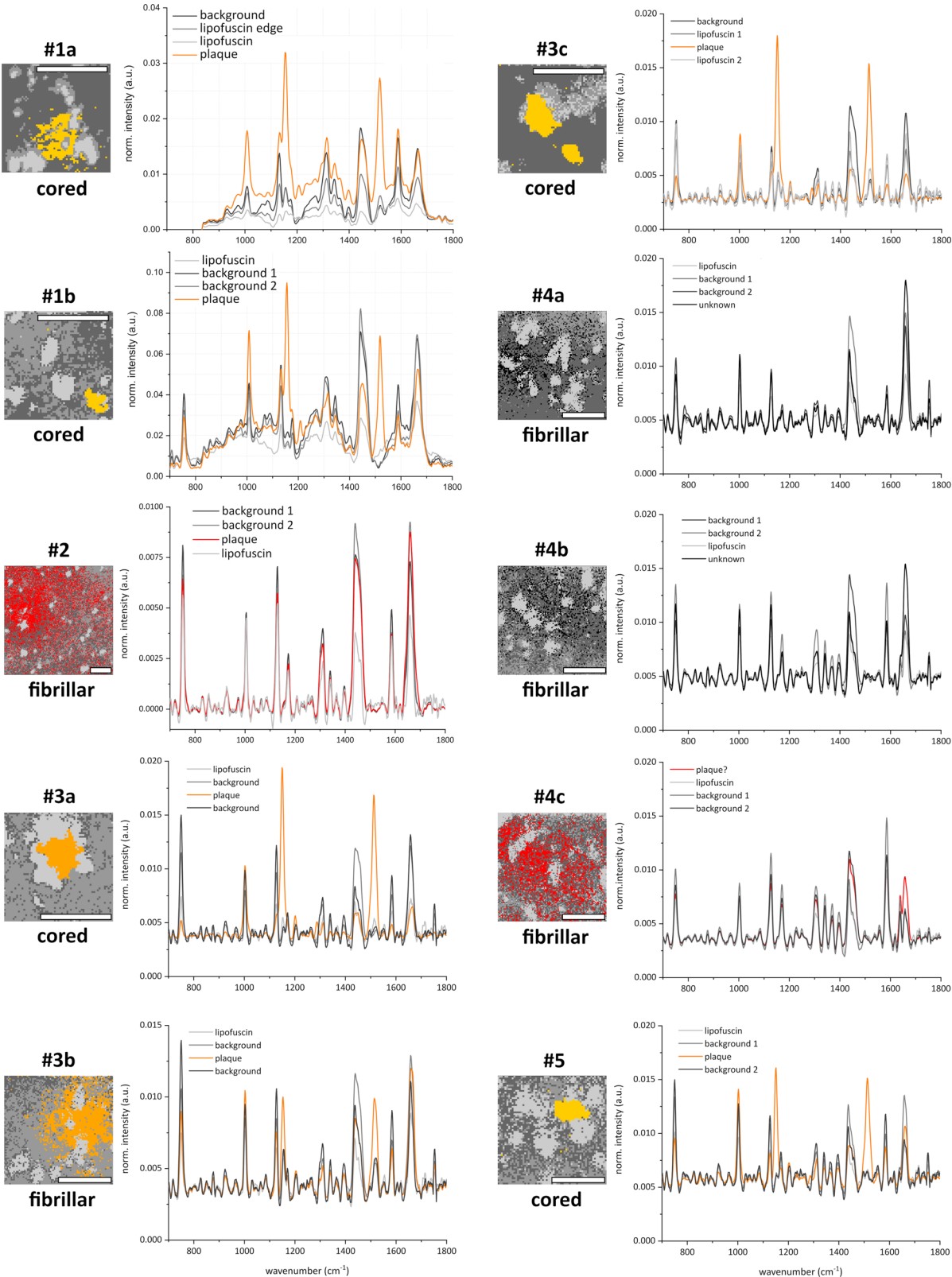

**Fig. 6 Four-cluster images of all plaques and their corresponding spectra after complete data processing.** The orange solid lines, obtained from the cored plaques and from fibrillar plaque #3b, exhibit a distinguishable spectrum with clear carotenoid-associated peaks. However, in plaques #2 and #4 (a–c) no carotenoid peaks were detectable. The red area, which largely overlaps with the later confirmed plaque location, corresponds to the spectrum (red line) on the right which is not associated with carotenoids. (The whole recorded spectra of #1a including the CH-stretch range can be found in the Supplementary Information, Fig. S11.) Scale bars: 40 μm.

found a factor of roughly 38 times increased efficiency for beta-carotene and 30 times for lutein when using a 532-nm source in comparison with the 785-nm laser (see Fig. S14 in the Supplementary Information, normalized to the solvent Raman intensity to correct for differences in laser power and other factors). Note that this enhancement factor at 532 nm may be different for other carotenoid compounds with a shifted absorption spectrum. In the absence of any substantial resonance enhancement, the carotenoid levels are too low to be detected with 785 nm excitation. Previous studies on carotenoids suggest to use a laser wavelength in the deep blue range to increase the resonance effect even further by better matching the excitation wavelength with the absorption band[59]. Beta-carotene and lutein are weakly fluorescent (see Fig. S15, in agreement with reference[60]) and their emission wavelength range matches that of the greenish auto-fluorescence of the plaque areas (under 488 nm excitation, see Fig. 4 and Fig. S5) quite well. However, it cannot be excluded that another plaque-associated compound, present at lower level but with a higher fluorescence quantum yield, is mainly responsible for this green emission. The molecular origin of the green fluorescence must be different from that of the blue emission observed under UV excitation by Thal et al.[37], indicating that there are at least two separate fluorescent compounds associated with plaques.

In Fig. S16, we compare the obtained Raman plaque spectra of #1a with Raman measurements of β-carotene dissolved in hexane. Beta-carotene is commonly used as reference sample for carotenoids while lutein, together with zeaxanthin, is one of the main carotenoids found in human brain tissue[61,62]. The three typical carotenoid peaks[63], approximately at 1007, 1156, and 1523/6 cm$^{-1}$, clearly match the extra peaks observed in plaques, see in Fig. S12 (bottom left) the averaged Raman spectra of plaque and non-plaque areas. This provides further proof that carotenoids are present at elevated levels within amyloid accumulations. The exact Raman peak positions may slightly vary, depending on the polyene chain length, side groups, and aggregation state of the specific carotenoid molecule[48,57]. In vitro and in vivo animal carotenoid studies suggest that carotenoids in general have antioxidant and anti-inflammatory effects[64,65] and lutein was observed to have a "strong suppressive impact on Aβ formations"[65–67]. Our results are in line with the observation that amyloid accumulation in AD brain is associated with an inflammatory process[68–72], and activated microglia are localized next to (the core of) plaques[28,70,73–75]. Also others reported carotenoid-associated Raman spectra in AD formalin-fixed human brain tissue[76] and in blood plasma[77] which supports the suggestion that the presence of carotenoids may be associated with an ongoing inflammatory process[68]. Carotenoids cannot be synthesized de novo by mammals and therefore must be obtained from diet[78]. Due to their lipophilic nature, they are expected to occur at a resting level in cell membranes and lipoprotein components[79]. Literature suggests that carotenoids, similar to microglia, are accumulating at sites of local inflammation (here, protein aggregation in AD), to fight oxidative damage, and reactive oxygen species (ROS)[64,79,80]. This might explain the elevated levels of carotenoids found at locations with high Aβ accumulations. As an indication that similar processes may have occurred in the AD cases of our study, we show microglia activity next to a core amyloid deposit, imaged in an adjacent section of sample #3c (see Fig. S17). Interestingly, retinal imaging studies in AD patients have shown that the macula pigment volume, consisting of carotenoids, is significantly lower compared to control patients[81].

**SRS microscopy.** Recently, Ji et al.[24] reported a protein peak shift towards the expected β-sheet peak (1670 cm$^{-1}$) in freshly frozen transgenic mouse brain tissue (with AD pathology) using SRS

micro-spectroscopy. Here, we applied a comparable SRS methodology to human AD samples, using the same tissue sections and areas of interest as in the previous sections. The results are given in Fig. 7 and for the control cases in Fig. S18 in the Supplementary Information. The plaques were raster scanned at different wavenumber settings across the protein peak in 3 cm$^{-1}$ steps (with smaller step sizes around the 1660 cm$^{-1}$ region, while plaque #1a was obtained using a larger step size). The data for the averaged background were taken within the yellow dashed circles, and the red dashed lines encompass the plaque areas. The data per plaque were normalized for a better comparison of the background and plaque curves. Please note that for the SRS measurements, the tissue sections on the microscope slides were not protected with a coverslip but only covered with deionized water. Therefore, the 20-μm-thick tissue might be slightly bloated and therefore appears non-flat under the microscope, which complicates the depth sensitive SRS measurements in terms of homogeneous field of view observation. That is the reason why we could not obtain SRS measurements from samples #3c, #4c and from control sample c#1b. In AD tissue sections #1b, #3a, and #3b we observe a strong and unique protein peak shift towards the expected β-sheet peak. The intensity of the background data has a maximum at 1659 cm$^{-1}$ while the plaque curve peaks at 1666 cm$^{-1}$. This is in agreement with the above-mentioned results from mouse brain tissue, where the examined plaque deposits can be described as ultra-dense and compact[24]. For tissue sections #1a, #4a, and #5, the protein peak shift is less pronounced compared to the former sections. Finally, for tissue section #2 and #4b the extensive spread of the amyloid fibrils with a presumably low molecular density hampers any clear spectroscopic distinction; the data appear very noisy and do not allow any unambiguous statement about a potential peak shift. In general, the results suggest that all cored plaques (#1a, #1b, #3a, and #5) that were found to contain carotenoid accumulations also display a protein peak shift when measured with SRS. In addition, a similar behavior can be observed for the amyloid deposit #3b, which we classified as fibrillar deposit based on its thioflavin-S staining pattern. As expected, the control cases do not show a protein peak shift within the imaged areas (as shown in Fig. S18 in the Supplementary Information). In summary, the obtained results are largely in agreement with published results on mice[24], with the peak shifts observed in the conventional Raman data (Fig. S9 in the Supplementary Information) and with our findings on carotenoid levels. Nevertheless, we are still far away from a label-free detection method for plaques in AD tissue via SRS since only a few of the plaques could be clearly identified, mostly the ones with a very high concentration of amyloid. Furthermore, it is worth to mention that our current data do not reveal a pronounced lipid halo surrounding the plaques as recently reported by several studies[19,24,82].

In summary, we sequentially applied multimodal and label-free imaging techniques on human AD brain tissue to identify different phenotypes of amyloid plaques. Mere auto-fluorescence imaging of snap-frozen, unstained AD gray matter tissue suggests the possibility of rapid visual assessment and localization of plaque pathology. The RGB camera of our full-field fluorescence microscope (rather than a single-wavelength bandpass filter) helped to distinguish the plaque's greenish auto-fluorescence from the stronger yellow-orange emission of the lipofuscin deposits. Thal et al.[37] reported that plaques areas show blue auto-fluorescence upon irradiation in the UV range. A comparative study should be carried out to find out which wavelength settings in excitation and emission would offer the best sensitivity and selectivity in label-free fluorescence microscopy applications for fast direct pathological examination of AD tissue. The label-free identification of amyloid plaques by its auto-fluorescence signal

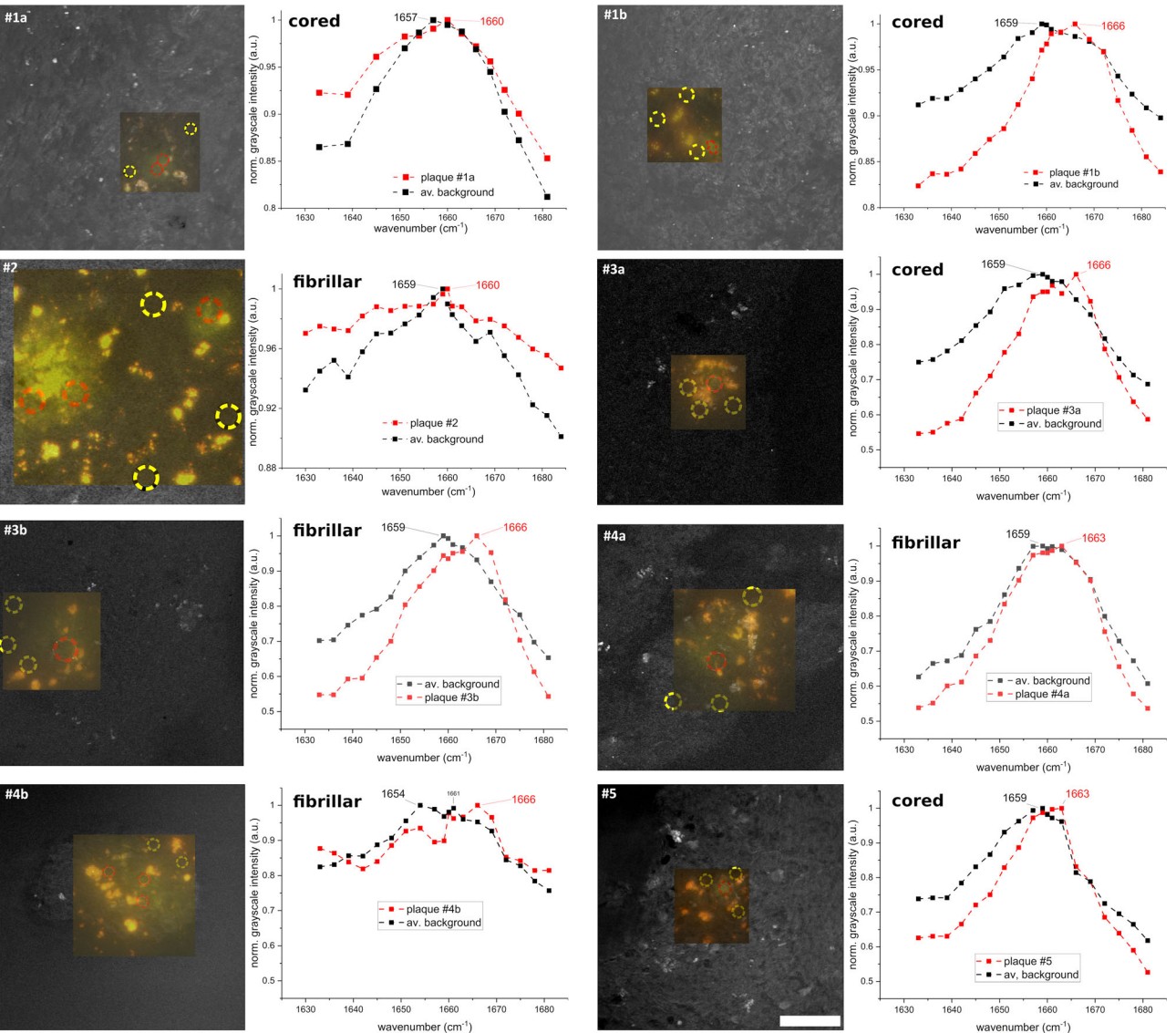

**Fig. 7 SRS images obtained while scanning across the protein peak for each plaque.** Auto-fluorescence images overlaid on SRS max. intensity projection images (@ 1666 cm$^{-1}$) with the plaque locations shown as green spots. Next to the images are the SRS response graphs of averaged sweeps across the protein peak per plaque. A clear shift from the "normal" 1659 cm$^{-1}$ protein peak towards the beta-sheet peak at 1666 cm$^{-1}$ is only observed in plaques featuring a dense cored plaque. This shift was not observed in the plaque #2 and less visible in #1a (which was measured with bigger step sizes). The yellow circles indicate areas where data were taken for the background and the red circles indicate the areas for plaques. Scale bar: 50 μm.

could provide a promising basis for in vivo (auto-) fluorescence brain and potential retinal imaging as the eye provides a clear optical window for human studies[83–85].

Despite its extended acquisition time, spontaneous Raman mapping under (pre-) resonance conditions adds vital and complementary information about the chemical specificity of the examined biomolecules. We identified the presence of carotenoids as a unique spectral feature associated with the location of (cored) plaques, which allowed us to differentiate areas with accumulations of amyloid protein from the surrounding tissue.

In the near future, advanced and rapid tissue imaging using SRS for amyloid plaques identification by monitoring and detecting a potential spectral shift from protein to β-sheet might have some potential pitfalls. In this preliminary study, only the cored amyloid plaques showed a peak shift, while fibrillar plaques showed no or only limited evidence of such a protein peak shift. Certainly, more research is needed to better understand the

underlying causes why the expected peak shift towards β-sheet deposits was not observable in all plaques. A major advantage of SRS over spontaneous Raman is of course the much higher mapping speed: even when the same area is imaged at tens of different wavenumber settings (as in this study), it is still about two orders of magnitude faster. Furthermore, SRS does not suffer from background fluorescence. Advanced strategies, e.g., via deep learning[86] might be able to enhance the information value of SRS imaging results. At the same time, future research should investigate the presence of carotenoids in AD tissue using advanced (VIS) SRS systems with at least one of the laser wavelengths closer to the carotenoids' absorption band[57]. Under such pre-resonance SRS conditions, we anticipate an improved sensitivity and selectivity of the carotenoids within AD tissue and an even higher mapping speed as compared to detecting a potential protein peak shift with NIR SRS[87].

Overall, the detection of carotenoids in amyloid accumulations indicates a neuroinflammatory response associated with

misfolded protein aggregations and its involvement in neuro-pathological structures in dementia. However, the implications of carotenoids co-localized with Aβ accumulations remains unanswered and further investigations are necessary to understand their role and why some plaques appear to have non-detectable carotenoid levels.

## Methods

**Brain tissue and tissue preparation.** Post-mortem brain tissue was obtained from the Netherlands Brain Bank (NBB), Netherlands Institute for Neuroscience (NIN), Amsterdam. All brain tissue was collected from donors with written informed consent for brain autopsy and the use of brain tissue and clinical information for research purposes. The brain donor program of the NBB was approved by the local medical ethics committee of the Vrije Universiteit Medical Center (Ref#2009/148). Neuropathological diagnosis was based on histochemical stainings including hematoxylin and eosin, Congo red staining, Bodian or Gallyas and Methenamine silver stainings and immunohistochemical stainings for amyloid-beta, p-tau, alpha-synuclein, and p62. These stainings were performed on formalin-fixed, paraffin embedded brain tissue of multiple brain regions, including the frontal cortex, temporal pole, superior parietal lobe, occipital pole, amygdala and the hippo-campus. Neuropathological diagnosis of AD was based on Braak stages for neu-rofibrillary tangles and amyloid[88], Thal phases for Aβ[89], as well as CERAD criteria for neuritic plaques[90]. We selected cases with a high score of AD pathology according to the NIA-AA criteria[91]. Based on post-mortem diagnosis, we excluded cases with other neurodegenerative diseases. Snap-frozen brain tissue sections (20 μm) from five selected cases were obtained from the frontal lobe. The area of interest was the gray matter region where plaques are expected and their presence was confirmed using histochemistry with thioflavin-S in the same section and immunohistochemistry against Aβ in sequential sections (see Fig. S1 for anti-Aβ staining in the Supplementary Information). Here, we report on the results of plaques imaged from five different AD brain donors. In total, ten amyloid plaques have been assessed. Patient demographics and plaque characteristics, based on their thioflavin-S stained pattern (cored vs. fibrillar), can be found in Table 1. The characterization of the five control sections from two donors can be found at the last rows in Table 1.

### Fluorescence microscopy

*Full-field fluorescence microscopy.* Auto-fluorescence images were acquired with a Leica DM2000 fluorescence microscope. The illumination source was a blue LED with a wavelength of about 470 nm. The color images were obtained for wavelengths above 500 nm using the attached Leica DFC450 C camera and the Leica application suite lite software. The same microscope and filter settings were used for the thioflavin-S-stained tissue sections after spontaneous Raman and SRS imaging.

*Confocal fluorescence scanning microscopy.* A confocal fluorescence scanning microscope (Nikon A1) with an A1-DUS spectral detector unit was used. The characterization of the emission curve of plaque locations was done using a 488 nm excitation source (close to the 470 nm LED source of the full-field fluorescence microscope) and a ×20 objective (0.8 NA). The power was kept constant at 1 mW with a pixel dwell time of 2.4 μs. The recorded emission ranged from 507 to 621 nm with a step size of 6 nm, resulting in 20 data points.

**Raman spectroscopy.** The spontaneous Raman measurements were recorded in mapping mode using a commercially available Raman spectrometer, Renishaw inVia, with an excitation wavelength of 532 nm. The attached Leica microscope was equipped with a ×50 objective (Leica HC PL FLUOTAR) with a NA of 0.8. The following settings were kept constant for each measured plaque. The laser power was set to 5% corresponding to 3 mW at the sample plane with a spot size of less than 1 μm. Per pixel, two or three accumulations were recorded with an exposure time of 1 s. The tissue sustained no detectable damage during the measurement. We recorded the fingerprint region from around 850–1850 cm$^{-1}$ with a resolution of approximately 4 cm$^{-1}$ FWHM. The total Raman spectra acquisition time per sample was up to 23 h, depending on the mapped area. Based on the prior acquired auto-fluorescence images, we mapped an area around the plaque location in 1 μm steps. To analyze and compare the resonance enhancement, additional measurements were done with an excitation source of 785 nm and the laser power set to 100%, measured as 82 mW at the sample plane.

**Stimulated Raman scattering microscopy.** For SRS measurements, we used an in-house built picosecond system as described in more detail in previous works[25,92,93]. In short, the Stokes beam ($\lambda = 1064$ nm, 80 MHz repetition rate, power 25 mW at the sample plane) was amplitude modulated at 3.6 MHz. The pump beam from the OPO was tuned from 901 to 907 nm to cover the amide-I protein peak in the fingerprint region (around 1660 cm$^{-1}$), with a power measured as 55 mW at the sample plane. The spectral resolution in this wavelength range is approximately 4 cm$^{-1}$. The tissue section was placed below the objective (Zeiss, C-Achroplan W, ×32, 0.85 NA, water immersion) and covered with deionized water.

During the recording the pixel dwell time per image was kept constant and set to 177.32 μs. Furthermore, the lock-in amplifier settings with a scale factor of +445.5 (a.u.), offset +0.100 mV, low-pass filter order 8, and a time constant of 25 μs were kept constant throughout all measurements. The obtained raster scanned images are an average of two measurements.

**Thioflavin-S staining.** Thioflavin-S staining was performed for confirmation of amyloid deposits, but only after auto-fluorescence imaging, spontaneous Raman imaging and SRS imaging. First, the tissue was fixed using formalin (4%) (10 min) to secure tissue on the microscope slide. Subsequently, the tissue sections were incubated with 1% thioflavin-S (Sigma-Aldrich) solution (demineralized water) for 10 min, followed by rinsing off excess thioflavin-S using 70% alcohol. Tissue sections were then washed using PBS and covered using Tris-buffered saline (TBS)/glycerol mounting medium, as described previously[17], and covered with a cover-slip. Thioflavin-S binds to Aβ that is stacked into a beta-pleated confirmation, also referred to as amyloid. This amyloid is present in cored and fibrillary plaques. Any possible non-fibrillary plaques[5] are likely not stained and hence the plaque might appear smaller than expected from well-known Aβ staining. Thioflavin-S staining is commonly used to identify plaque in tissue[20,24,94].

**Data processing.** The spectral data recorded from the Raman spectrometer were processed entirely using OCTAVVS (version 0.0.28)[95], a recently developed open chemometrics toolkit consisting of three steps: (data) pre-processing, multivariate curve resolution-alternating least squares (MCR-ALS) and clustering. In the pre-processing step, the spectra were de-noised (Savitzky-Golay filter with a window size of nine data points corresponding to approximately 9.4 cm$^{-1}$ and a third-order polynomial), baseline corrected (using the asymmetric reweighted penalized least-squares (arPLS) algorithm) and normalized (area under the curve). In step two, a single value decomposition (SVD) for eight components was performed (with 5% noise allowed). Afterward, the MCR-ALS was started using the non-negatively constrained least-squares iteration algorithm to reach a tolerance below 0.1%. The final clustering step was performed based on the pure spectra calculated by MCR-ALS. Eventually, the computed spectra were clustered in four groups using area normalization and the Mini Batch K-Means algorithm. For a more detailed description of these processing algorithms we refer to the paper of the OCTAVVS developers[96].

**Reporting summary.** Further information on research design is available in the Nature Research Reporting Summary linked to this article.

## Data availability

The data that support the findings of this study are available from the corresponding author upon reasonable request. The source data of the main files is given in Supplementary Data 1.

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

## Acknowledgements
The authors would like to thank Tjado H. J. Morrema for technical assistance and C. Troein for his software support. This research is supported by the Dutch Technology Foundation STW (grant number 13935, I-READ), which is part of the Netherlands Organization of Scientific Research (NWO), and which is partly founded by the Ministry of Economic Affairs.

## Author contributions
B.L. performed the imaging tasks, analyzed the data, compiled the results, and wrote most of the paper. B.D.C.B. provided the brain tissue and assistance with the neuropathological characterization, performed the emission spectra measurements, and wrote part of the paper. S.R.V. assisted with the emission spectra measurements and performed the carotenoid compounds measurements. L.Z. supported the SRS imaging task and provided technical support. J.J.M.H. prepared the human samples, wrote the sample preparation and immunohistochemical paragraphs, and performed the stainings. F.A. performed the carotenoid measurements, provided the chemical analysis, and was heavenly involved in the manuscript editing. J.F.d.B. conceived and designed the work and edited the text. All co-authors discussed and commented on the manuscript.

## Competing interests
The authors declare the following competing interests. Heidelberg Engineering was part of the consortium funding the research (I-READ).
