## [Peer Review File · Communications Biology]

Reviewers' comments:

Reviewer #1 (Remarks to the Author):

Lochocki et al. report detection of amyloid plaques by label-free fluorescence and Raman imaging in snap-frozen frontal cortex tissue from Alzheimer's disease (AD) cases. They analyzed single plaques from sections of two AD cases. Controls were not included. Autofluorescence was observed by excitation at 470 nm and detection in the spectrum of >500 nm. For Raman imaging the authors chose an excitation wavelength of 532 nm or 785 nm. The Raman spectrometer recorded around the fingerprint region from 850 – 1850 cm^{-1} .

With these settings a very weak greenish autofluorescence of amyloid plaques was shown. Mainly the amyloid plaque cores were positive. The detection of amyloid was confirmed by thioflavin S staining of the plaques. Such plaques, identified by their autofluorescence, were observed with Raman imaging showing peaks of 1518 cm^{-1} and 1154 cm^{-1} , indicative for carotenoid compounds. The authors conclude that amyloid autofluorescence is a novel option for label-free detection of amyloid plaques and that Raman imaging allows the detection of carotenoids in amyloid plaques. The identification of carotenoids in amyloid plaques by Raman imaging is indeed new. Autofluorescence of amyloid plaques is not novel. Since 2002 it is known that amyloid plaques have autofluorescence properties that are best visualized when exciting at 360-370 nm and detecting at >420 nm.¹ Therefore, I wonder why the authors did not test the entire spectrum of possible excitation wavelengths when searching for autofluorescence properties of amyloid plaques. Moreover, the use of only two cases without controls leaves the question of reproducibility and specificity of the findings.

Specific points:

1. Autofluorescence of amyloid plaques is not a novel finding. The authors should cite and discuss the appropriate literature and include alternative excitation strategies reported in literature for comparison with their findings.
2. The number of cases used in this study needs to be significantly increased (e.g. $n = 10$ AD and 10 controls) to deliver more reliable results. It is essential to clarify whether the findings are specific findings related only to single cases or whether they are relevant for AD in general.
3. The authors did not include control cases.
4. The authors measured only single plaques. It is not clear whether they confirmed their findings in a higher number of plaques per case. I would expect that at least 20 plaques should be measured per case.
5. The authors interpret the presence of carotenoids in amyloid plaques as indicative for neuroinflammatory response to misfolded protein accumulation. However, they do not show whether the carotenoid-containing plaques are associated with microglia response or the presence of other neuroinflammation markers.

Reference:

1 Thal, D. R., Ghebremedhin, E., Haass, C. & Schultz, C. UV light-induced autofluorescence of full-length A β -protein deposits in the human brain. *Clin Neuropathol* 21, 35-40 (2002).

Reviewer #2 (Remarks to the Author):

In this manuscript entitled "multimodal, label-free fluorescence and Raman imaging of amyloid deposits in snap-frozen Alzheimer's disease human brain tissue", the authors presented a comprehensive study of various optical properties of amyloid fibrils and deposits in human AD frozen tissue sections. Although fluorescence and Raman spectroscopy and microscopy of A β -plaques have been investigated in previous studies, the current work discovered strong autofluorescence and carotenoid Raman spectra associated with amyloid plaques. These findings are certainly valuable to the researches in the field. The authors also pointed out the complication of Raman/SRS identification of misfolded proteins in the plaques. The manuscript was well written with combined data using different types of techniques, but the major issue seems to be that the underlying mechanisms remain untouched, such as the source of the greenish autofluorescence of the plaques. I would suggest the authors at least give some efforts and hypothesize the potential cause of these phenomena. For instance, the spectra of the autofluorescence may be taken and

analyzed.

Reviewer #3 (Remarks to the Author):

In this manuscript, Lochocki et.al. presented a multimodal imaging of amyloid deposits in human Alzheimer tissue. Three different modalities – autofluorescence, Raman, and stimulated scattering Raman microscopy – are sequentially used to map the amyloid plaques that are confirmed with Thioflavin-S staining. A few features that are obtained are quite interesting and could be of diagnostic value, including the green autofluorescence and the presence of carotenoid in certain plaques. However, simply putting three different modalities together (not on the same platform) do not necessarily offer an improved understanding of amyloid aggregation. In particular, SRS shown here does not seem to add much value, especially considering that Raman images are already obtained. There are a few major improvements that can be implemented before publication.

1. Autofluorescence of amyloid plaque in human patient has been demonstrated (Clin Neuropathology, 2002;21(1):35-40). Detailed characterization of Amyloid-beta plaque has also been shown (Amyloid Proteins. Methods in Molecular Biology, vol 1779). It would be interesting if the fluorescence is better characterized in terms of spectral properties besides description of the amyloid plaque as greenish structures on top of lipofusion orange fluorescence background. It seems that a simple bandpass filter would be able to separate the amyloid plaque better from the lipofusion. The faint green is difficult to visualize if not highlighted with boxes and arrows. Especially for patient 2, many green plaques overlap with orange, which are similar to some of the green in patient 1 not highlighted by the authors.
2. It would be important to have some quantitative analysis done on the correlation of autofluorescence/Raman and staining to support the use of such features for pathological examination. Currently, only a few selected areas with image comparisons are shown, which can be heavily biased by the choice of highlighted area.
3. The Raman features of carotenoids are quite interesting and more convincing. It is surprising that only patient 1 shows these features but not patient 2. It is possible that they have different plaques. More patients would be better but it is understandable that there are limitation to how many samples can be studied. Sample 1a and 1b are pretty much the same for most features. It is good to show consistency of the result, but why not do that for patient 2? What are the plaque differences between the two patients? One is said to have cored amyloid while the other is fibrillar amyloid. Are they based on pathological grading or simply thioflavin staining pattern? The use of fibrillar amyloid in the manuscript is confusing. While patient 1 is designated to have cored amyloid plaque but description of all cases in Figure 2a and 2b says autofluorescence and thioflavin staining highlight fibrillar amyloid deposits (line 209, 216, and others).
4. The SRS images do not seem to provide any information and can be taken out from the paper without affecting the results. Moreover, the SRS imaging results are not consistent with previous SRS and CARS study. It is possibly due to sample difference. However, it is also possible that the SRS imaging was not done properly with the best SNR achievable. For example, the SRS spectra showed very large background signal (>80%, Figure 7) with noisy and inconsistent peak features, which undermines the validity of subsequent data analysis and claims.

Other comments:

1. In Figure 2, the green box is confusing. It seems to highlight the zoom-in area, but when looked closely, the yellow box and surround areas are actually zoomed-in, not the green box.
2. What is the color scale for Figure 4? From the raman spectra, it looks like the plaque should have higher Raman intensity, which is not the case in the actual total intensity image or protein image. Is that because the Raman spectra are individually scaled?
3. In patient 2, common Raman peaks found in plaque are all present in surrounding tissue. The author suggested that "other spectral peaks are present but to a lesser extent in the spectra of plaques #1a and #1b". This seems to imply other peaks can be used to differentiate the plaque from normal tissue, but exactly which peaks can do that and why do they highlight plaque are not clarified. The image shown also did not exactly highlight the plaque.
4. The resonance enhancement is important for extracting carotenoid features. Using 532nm for carotenoid detection is already widely recognized in the literature. There is no need to discuss in

great details about the benefit of 532 nm excitation for carotenoid detection. Figure 6 is useful for proving the identify of the observed peaks, but it is more suitable as a supplementary figure, which allows the manuscript to flow better (showing the identify first and then make claims about how it can be used for imaging plaque location).

First, we would like to thank the referees for their careful and thorough reading of our manuscript and for their constructive suggestions, which helped us to improve the quality of the submitted manuscript. In spite of restricted access to the laboratory, we carried out additional measurements: characterizing additional AD cases, performing the same analyses on control tissues, measuring reference spectra for a second carotenoid compound (lutein) and looking for activated microglia. Furthermore, we added corrections, clarifications (where needed) and additional figures (mainly in the Supplementary Information) which hopefully meet your expectations.

What follows is a point-by-point response to the comments made by the referees.

Reviewers' comments:

Reviewer #1 (Remarks to the Author):

Lochocki et al. report detection of amyloid plaques by label-free fluorescence and Raman imaging in snap-frozen frontal cortex tissue from Alzheimer's disease (AD) cases. They analyzed single plaques from sections of two AD cases. Controls were not included. Autofluorescence was observed by excitation at 470 nm and detection in the spectrum of >500 nm. For Raman imaging the authors chose an excitation wavelength of 532 nm or 785 nm. The Raman spectrometer recorded around the fingerprint region from 850 – 1850 cm^{-1} . With these setting a very weak greenish autofluorescence of amyloid plaques was shown. Mainly the amyloid plaque cores were positive. The detection of amyloid was confirmed by thioflavin S staining of the plaques. Such plaques, identified by its autofluorescence were observed with Raman imaging showing peaks of 1518 cm^{-1} and 1154 cm^{-1} , indicative for carotenoid compounds. The authors conclude that amyloid autofluorescence is a novel option for label free detection of amyloid plaques and that Raman imaging allows the detection of carotenoids in amyloid plaques. The identification of carotenoids in amyloid plaques by Raman imaging is indeed new. Autofluorescence of amyloid plaques is not novel. Since 2002 it is known that amyloid plaques have autofluorescence properties that are best visualized when exciting at 360-370 nm and detecting at >420 nm.¹ Therefore, I wonder why the authors did not test the entire spectrum of possible excitation wavelengths when searching for autofluorescence properties of amyloid plaques. Moreover, the use of only two cases without controls leaves the question of reproducibility and specificity of the findings.

We thank the reviewer for these remarks; the comments on plaques autofluorescence, the number of cases examined and control samples will be discussed below under specific points 1, 2 and 3, respectively.

Specific points:

1. Autofluorescence of amyloid plaques is not a novel finding. The authors should cite and discuss the appropriate literature and include alternative excitation strategies reported in literature for comparison with their findings.

We would like to thank the reviewer for that comment. We are aware that auto-fluorescence of amyloid beta plaques is not a novel finding, as we also do not state that, and we had included recent literature. However, we unfortunately missed the mentioned publication in the manuscript and therefore, following the suggestion of the reviewer, we added the suggested reference (ref: #46) and added it to the discussion of auto-fluorescence in Section 3.1, to Section 3.2.1 and to the Conclusion. (Thal, D. R., Ghebremedhin, E., Haass, C. & Schultz, C. UV light-induced autofluorescence of full-length Aβ-protein deposits in the human brain. *Clin Neuropathol* 21, 35-40 (2002)). However, we would like to point out that the blue emission observed by Thal et al. must be due to a different type of compound than the greenish fluorescence and the Raman signatures described in this work.

2. The number of cases used in this study needs to be significantly increased (e.g. n = 10 AD and 10 controls) to deliver more reliable results. It is essential to clarify whether the findings are specific findings related only to single cases or whether they are relevant for AD in general.

We agree with the reviewer that the number of cases should be increased. Therefore, we did further measurements on different AD cases and added measurements done on tissues from two control patients. We increased the number of AD cases to 5 different patients, resulting in 10 different plaques which were characterized. The additional measurements support the findings of the original submitted manuscript.

3. The authors did not include control cases.

Indeed, in the original manuscript the non-plaque areas from the same AD brain slices were used for comparison. As suggested by the reviewer and to improve the manuscript, we added 5 control measurements from two control cases and followed the same imaging procedures (auto-fluorescence, Raman, SRS, thio-S fluorescence). Please see the new Table S1 and new Figures S4 (auto-fluorescence), S7 (Raman maps), S9 (Raman spectra) and S17 (SRS) for further characterization of the control cases. The results obtained for these samples fully agree with those of the non-plaque areas of AD patients.

4. The authors measured only single plaques. It is not clear whether they confirmed their findings in a higher number of plaques per case. I would expect that at least 20 plaques should be measured per case.

Following this reviewer's suggestions, we already added several samples from extra AD cases and controls. Measuring also a larger number of plaques per patient could indeed strengthen the manuscript even further, but in combination with Raman mapping and SRS, we do not think that that would be a feasible approach, timewise. Especially not if we also, as the reviewer suggested, increase the number of cases to 10. That would be more than 400 sets of measurements (including the control cases) and therefore for logistics reasons the mapping experiments were limited to 1 to 3 plaques per patient. One of the other reviewers actually acknowledges (point 3) that there is a limit to the number of samples that can be fully characterized (Rev. 3 point 3). However, during the research we looked at many tissue sections and their auto-fluorescence images, and in general, we noticed that all green fluorescence spots were positively stained by thio-s afterwards (see e.g, Figure 2 and 3). However, we agree that it would be very interesting

to see whether all plaques one could find in a single tissue section express the same kind of autofluorescence (and carotenoid deposits).

5. The authors interpret the presence of carotenoids in amyloid plaques as indicative for neuroinflammatory response to misfolded protein accumulation. However, they do not show whether the carotenoid-containing plaques are associated with microglia response or the presence of other neuroinflammation markers.

Following this remark, we tried to additionally stain the already stained thioflavin-S tissue for microglia activity. Unfortunately, due to technical reasons, our staining for the microglia marker iba1 on the label-free imaged tissue failed therefore we cannot provide any direct link between microglia activity and carotenoid-containing plaques in the same tissue slice. However, in literature there are many examples of microglial activation within plaques. See for example, recent publications by one of the co-authors (Boon et al., *Journal of Neuroinflammation*, 2018 / Boon et al., *Acta Neuropathologica*, 2020).

In this manuscript we did not directly confirm microglia in the area in which we detected a carotenoid signal. However, we did perform a staining for iba1 and thio-S on the sequential section of the label-free imaged section containing dense cored plaques. In this staining (Supplemental Figure S16) one can clearly see microglia surrounding the thio-S positive core of a dense-cored plaque. It fully supports our conclusion that carotenoids indicate a neuroinflammatory response but further investigations are necessary.

Reviewer #2 (Remarks to the Author):

In this manuscript entitled “multimodal, label-free fluorescence and Raman imaging of amyloid deposits in snap-frozen Alzheimer’s disease human brain tissue”, the authors presented a comprehensive study of various optical properties of amyloid fibrils and deposits in human AD frozen tissue sections. Although fluorescence and Raman spectroscopy and microscopy of A-beta plaques have been investigated in previous studies, the current work discovered strong autofluorescence and carotenoids Raman spectra associated with amyloid plaques. These findings are certainly valuable to the researches in the field. The authors also pointed out the complication of Raman/SRS identification of misfolded proteins in the plaques. The manuscript was well written with combined data using different types of techniques, but the major issue seems to be that the underlying mechanisms remain untouched, such as the source of the greenish autofluorescence of the plaques. I would suggest the authors at least give some efforts and hypothesize the potential cause of these phenomena. For instance, the spectra of the autofluorescence may be taken and analyzed

We thank the reviewer for these comment and suggestions. We are happy to confirm that by carrying out additional measurements on a different instrument we managed to determine the emission spectra. We obtained additional amyloid deposit sections from the same cases (as #3 and #4) and measured their emission response curves using a confocal fluorescence microscope in spectral detection mode. The

results are discussed in the paragraph above Figure 4 and presented in Figure 4 and S5. We discuss the possible origin of the green fluorescence in Results Section 3.2.1

Reviewer #3 (Remarks to the Author):

In this manuscript, Lochocki et.al. presented a multimodal imaging of amyloid deposits in human Alzheimer tissue. Three different modalities – autofluorescence, Raman, and stimulated scattering Raman microscopy – are sequentially used to map the amyloid plaques that are confirmed with, Thioflavin-S staining. A few features that are obtained are quite interesting and could be of diagnostic value, including the green autofluorescence and the presence of carotenoid in certain plaques. However, simply putting three different modalities together (not on the same platform) do not necessarily offer an improved understanding of amyloid aggregation. In particular, SRS shown here does not seem to add much value, especially considering that Raman images are already obtained. There are a few major improvements that can be implemented before publication.

1. Autofluorescence of amyloid plaque in human patient has been demonstrated (Clin Neuropathology, 2002;21(1):35-40). Detailed characterization of Amyloid-beta plaque has also been shown (Amyloid Proteins. Methods in Molecular Biology, vol 1779). It would be interesting if the fluorescence is better characterized in terms of spectral properties besides description of the amyloid plaque as greenish structures on top of lipofusion orange fluorescence background. It seems that a simple bandpass filter would be able to separate the amyloid plaque better from the lipofusion. The faint green is difficult to visualize if not highlighted with boxes and arrows. Especially for patient 2, many green plaques overlap with orange, which are similar to some of the green in patient 1 not highlighted by the authors.

Following the reviewer's suggestion, we added the recommended reference (see also response to reviewer #1) and included it in sections 3.1 and 3.2.1 where we discuss the fluorescence characterization of amyloid-beta plaques. Furthermore, we determined the emission spectrum of amyloid deposit autofluorescence using a 488 nm excitation source. We added these results to section 3.1 and show the recorded emission curves of plaques and of lipofuscin in Figure 4 and S5.

We do not think that a simple bandpass filter would enable us to distinguish amyloid from lipofuscin since lipofuscin has a strong and broad emission response (as also discussed in our previous publication; Lochocki et al., 2020) and would still overlap with the amyloid auto-fluorescence in the green channel (see also the spectra of figures 4 and S5). In fact, we believe that a dual-band ratiometric analysis of the green vs. yellow-orange emission (as done by the RGB camera employed) is crucial to distinguishing the plaque areas from lipofuscin deposits and this was added to the Conclusion.

Indeed, we agree with the reviewer that on paper the captured images do not show very clear the greenish fluorescence and they appear faint. However, under the microscope, they are clearly visible and easy to distinguish by eye from the surroundings. This observation was added to Section 3.1

2. It would be important to have some quantitative analysis done on the correlation of autofluorescence/Raman and staining to support the use of such features for pathological examination. Currently, only a few selected areas with image comparisons are shown, which can be heavily biased by the choice of highlighted area.

We thank the referee for that comment. In a sense, our choice of area is indeed biased since we make a deliberate choice on what we want to image. It should be clear that we were mainly interested in the areas where we observed green auto-fluorescence and not in any random location. Especially conventional Raman mapping would be too slow to look at a large number of (mostly negative) randomly selected sites. However, for the newly added control cases, we picked random locations within the gray matter of the tissue (as we do with the AD cases). Quantifying the results in terms of sensitivity and specificity percentages would require many more samples and would be beyond the scope of this work.

In this revised manuscript we added more cases, and it becomes more evident that at least all examined dense cored amyloid deposits contain significant levels of carotenoids. In contrast, we were not able to detect carotenoids in fibrillar amyloid deposits (exception #3b). This is an interesting finding and might add to the hypothesis that neuroinflammation is involved in the formation of different types of plaques. However, we agree with the reviewer that more research is needed to understand the reasoning behind that.

3. The Raman features of carotenoids are quite interesting and more convincing. It is surprising that only patient 1 shows these features but not patient 2. It is possible that they have different plaques. More patients would be better but it is understandable that there are limitation to how many samples can be studied. Sample 1a and 1b are pretty much the same for most features. It is good to show consistency of the result, but why not do that for patient 2? What are the plaque differences between the two patients? One is said to have cored amyloid while the other is fibrillar amyloid. Are they based on pathological grading or simply thioflavin staining pattern? The use of fibrillar amyloid in the manuscript is confusing. While patient 1 is designated to have cored amyloid plaque but description of all cases in Figure 2a and 2b says autofluorescence and thioflavin staining highlight fibrillar amyloid deposits (line 209, 216, and others).

We would like to thank the reviewer for this comment and agree that the two different uses of the word “fibrillar” can cause confusion. In the revised manuscript this was rephrased, and we mention more clearly that the plaques are of different types. We differentiate between cored and fibrillar amyloid deposits, and the additional tissue sections analyzed for this manuscript revision contain both types and they support the findings obtained for cases #1 and #2. The amyloid plaques that show different Raman signatures might be different types of plaque meaning a plaque of different origin and formation. (For different plaque phases, please see references #27. Interestingly with the increased number of scanned plaques, we observed that all cored plaques had carotenoid signals, whereas of the scanned fibrillar plaques only 1 plaque showed carotenoid signal. That 1 fibrillar plaque was found in a case with mostly dense cored plaques and thus we suggest that this plaque might be in a different stage. The other fibrillar plaques were

seen in cases where most plaques were of the fibrillar plaque type (Boon et al., *Acta Neuropathologica*, 2020). The difference in carotenoid signal between the cored and fibrillar plaque types may suggest that neuroinflammation in those 2 types differs and thus pathophysiology of plaque formation differs.

As stated in the methods section 2.1, the AD cases are graded according to their pathological immunohistochemical findings in Abeta immunostaining (and not on the thioflavin-S staining) prior to any imaging modality. However, the distinction if a plaque is a cored or fibrillar amyloid deposit, was made based on the thio-S stained pattern.

4. The SRS images do not seem to provide any information and can be taken out from the paper without affecting the results. Moreover, the SRS imaging results are not consistent with previous SRS and CARS study. It is possibly due to sample difference. However, it is also possible that the SRS imaging was not done properly with the best SNR achievable. For example, the SRS spectra showed very large background signal (>80%, Figure 7) with noisy and inconsistent peak features, which undermines the validity of subsequent data analysis and claims.

Thanks to the reviewer for this observation; it is true that the SRS-derived spectra of the original manuscript were a bit noisy. Following a thorough overhaul of the setup, stronger and more stable SRS signals were obtained. As we could not repeat the SRS measurement on the already stained thio-S tissue sections we added the results of more SRS measurements with improved SNR (as can be seen in the new Figure 7 and S17) acquired from the additional AD and healthy control cases. The new SRS results largely agree with the previously recorded SRS images.

One reason why the quality of the SRS graphs is not ideal is the sample preparation, which is not optimized for SRS since the sample has to be suitable for various imaging techniques and afterwards accessible for staining. Image quality improvements could for instance be achieved by flattening the sample with a cover slip if we would only focus on SRS imaging. (In addition, the fingerprint region is usually more challenging to image compared to e.g the CH-stretch region).

Furthermore, the SRS results may not necessarily be consistent with previous studies since we are presenting novel SRS results taken on human AD tissue while previous studies were done on transgenic mice tissue. Secondly, our results are suggesting that a protein peak shift can be observed (as in transgenic AD mice tissue) but only when looking at (dense) cored plaques. Thirdly, the conventional Raman measurements exhibit new Raman bands within plaques (carotenoids) while the protein shift was difficult to observe (see Figure S6). In contrast, SRS could highlight the protein shift but not the carotenoid bands (due to the non-resonance wavelengths used in the SRS system). Hence, the combination of conventional Raman and SRS helps to add value for future research by showing potential differences of both techniques and helps to guide researchers when similar measurements might be considered.

In summary, we agree with the reviewer that the SNR of our system was not the best and for further measurements improvements were made (see Fig. 7). Especially with the combined results of Raman mapping, we are convinced that the SRS measurements are useful to help distinguish between plaque types.

Other comments:

1. In Figure 2, the green box is confusing. It seems to highlight the zoom-in area, but when looked closely, the yellow box and surround areas are actually zoomed-in, not the green box.

Thanks for that observation. We followed the reviewer's suggestion and removed the green boxes.

2. What is the color scale for Figure 4? From the raman spectra, it looks like the plaque should have higher Raman intensity, which is not the case in the actual total intensity image or protein image. Is that because the Raman spectra are individually scaled?

That is actually a nice observation by the reviewer. Thanks for noticing. The color scale (now Fig. 5) is from blue (low) to green to yellow (high) and represents the values from the spectral graphs (as shown in Fig.6). Indeed, the first plaque should have a higher intensity if one is looking at the spectra of Fig. 6. In this particular case however, the total intensity image was calculated from the full spectrum (including the strong C-H stretch emission, see Supplementary Information Fig. S10), and therefore the total intensity image looks incorrect if based on the assumption that only the spectrum shown in Fig. 6 was taken. To avoid further confusion, we limited the wavenumber range of all intensity images to the fingerprint region from 700 to 1800 cm^{-1} (except for case #1a where it is from 837 to 1800 cm^{-1} because we do not have measurements below 837 cm^{-1}) and re-calculated the total intensity images for the revised manuscript.

3. In patient 2, common Raman peaks found in plaque are all present in surrounding tissue. The author suggested that "other spectral peaks are present but to a lesser extent in the spectra of plaques #1a and #1b". This seems to imply other peaks can be used to differentiate the plaque from normal tissue, but exactly which peaks can do that and why do they highlight plaque are not clarified. The image shown also did not exactly highlight the plaque.

Thanks for the comment and apologies for the confusion. We simply meant the common tissue peaks of 1003, 1445 and 1666 cm^{-1} , which are always to be expected in biological human tissue, are present in both AD (Fig. 6) and control (Fig. S9), but are not specific for plaques. Further we state that apart from these common tissue peaks, we obtained other peaks after the MCR-ALS data processing, which were equally present in all samples (the mentioned 760, 1130, 1311, 1586 cm^{-1} peaks). Since these peaks could be found in all clustered spectra, they cannot be used for any differentiation either. We added a sentence to clarify that in the manuscript (end of Section 3.1).

4. The resonance enhancement is important for extracting carotenoid features. Using 532nm for carotenoid detection is already widely recognized in the literature. There is no need to discuss in great details about the benefit of 532 nm excitation for carotenoid detection. Figure 6 is useful for proving the identify of the observed peaks, but it is more suitable as a supplementary figure, which allows the manuscript to flow better (showing the identify first and then make claims about how it can be used for imaging plaque location).

We appreciate the comment, followed the suggestion of the reviewer and moved (previous) Figure 6 to the Supplementary Information, shown now as Figure S12.

REVIEWERS' COMMENTS:

Reviewer #1 (Remarks to the Author):

This is the revised version of a previously submitted manuscript. The number of cases is now significantly increased. Two control cases were included in the measurements. Other studies and wavelengths paradigm are now sufficiently discussed. By doing so, the authors addressed most of my points. Only few minor points remain to be adapted.

Specific points:

1. Please include the control cases in table 1.
2. Please state that this project has been approved by the local ethical committee.
3. Please provide the numbers of precise plaques investigated for each test and provide information about the reproducibility of the findings.
4. The authors interpret the presence of carotenoids in amyloid plaques as indicative for neuroinflammatory response to misfolded protein accumulation. However, they do not show whether the carotenoid-containing plaques are associated with microglia response or the presence of other neuroinflammation markers. I agree that other studies have shown that already but it will be essential to include such information in the discussion when talking about a neuroinflammatory response and carotenoids. Which cells produce the carotenoids? Under which neuroinflammatory conditions? Were these conditions observed in plaques (literature study would be enough)?

Reviewer #2 (Remarks to the Author):

The authors have carefully addressed my previous concerns, and I would like to recommend its publication in Communications Biology.

Reviewer #3 (Remarks to the Author):

The authors have largely addressed my concerns. One remaining issue is that I still do not think SRS provides information that is not already available with the other methods presented. In the rebuttal letter, the authors say "we are convinced that the SRS measurements are useful to help distinguish between plaque types". Yet, for the results shown in Figure 7, some core plaque has a shift, while others not so clear (#5); fibrillar plaque also shows inconsistent behavior (#3b has shift but not others). Coupled with the fact that data is noisy with broad features that are not from Raman contribution, I do not see how it supports the claim that it agrees with carotenoid levels (separates core from fibrillar) or aid differentiation.

We would like to thank all reviewers for their careful and thorough reading of our revised manuscript and their additional constructive suggestions which helped us to further improve the quality of the manuscript. We added clarifications, an additional figure (new) S9 in the Supplementary Information and an extra table (new) ST1 which hopefully meet your expectations.

What follows is a point-by-point response to the comments made by the referees.

Reviewer #1 (Remarks to the Author):

This is the revised version of a previously submitted manuscript. The number of cases is now significantly increased. Two control cases were included in the measurements. Other studies and wavelengths paradigm are now sufficiently discussed. By doing so, the authors addressed most of my points. Only few minor points remain to be adapted.

We are happy that we could clarify and address most of the reviewers' questions and points.

Specific points:

1. Please include the control cases in table 1.

Following the suggestion of the reviewer we merged the AD and control case tables and now present all investigated cases in Table 1 in the main manuscript.

2. Please state that this project has been approved by the local ethical committee.

We added the approval of the ethics committee in paragraph 2.1.

3. Please provide the numbers of precise plaques investigated for each test and provide information about the reproducibility of the findings.

In all AD cases we could confirm that the green auto-fluorescence patches were Thio-S positive (page 13, Figure 3). All cored plaques were found to contain carotenoids (5 out of 5). For fibrillar plaques carotenoids were found in 1 out of 5 cases (page 18, Figure 5). In none of the control samples carotenoids were found. We added a table into the Supplementary Information (Table ST1) showing which samples were investigated in each test.

4. The authors interpret the presence of carotenoids in amyloid plaques as indicative for neuroinflammatory response to misfolded protein accumulation. However, they do not show whether the carotenoid-containing plaques are associated with microglia response or the presence of other neuroinflammation markers. I agree that other studies have shown that already but it will be essential to include such information in the discussion when talking about a neuroinflammatory response and carotenoids. Which cells produce the carotenoids? Under which neuroinflammatory conditions? Were these conditions observed in plaques (literature study would be enough)?

We added in paragraph 3.2.1 a brief section on our understanding of the neuroinflammatory response and added appropriate references. Mammalian cells cannot produce carotenoids, but carotenoids (obtained earlier from the diet) can be transported to specific locations to fight inflammation. However, as we state in our conclusion "the implication of carotenoids co-localized with A β accumulations remains unanswered and further investigations are necessary to understand their role".

Reviewer #2 (Remarks to the Author):

The authors have carefully addressed my previous concerns, and I would like to recommend its publication in Communications Biology.

We appreciate the reviewer's recommendation.

Reviewer #3 (Remarks to the Author):

The authors have largely addressed my concerns. One remaining issue is that I still do not think SRS provides information that is not already available with the other methods presented. In the rebuttal letter, the authors say "we are convinced that the SRS measurements are useful to help distinguish between plaque types". Yet, for the results shown in Figure 7, some core plaque has a shift, while others not so clear (#5); fibrillar plaque also shows inconsistent behavior (#3b has shift but not others). Coupled with the fact that data is noisy with broad features that are not from Raman contribution, I do not see how it supports the claim that it agrees with carotenoid levels (separates core from fibrillar) or aid differentiation.

The reviewer is correct insofar that the shift of the protein Amide-I peak to higher wavenumbers that we observed with SRS was "already available with the other methods presented". Indeed, also the spontaneous Raman spectra of the plaque areas showed a similar shift for most of the cored plaque cases (see new Figure S9). For most of the fibrillar plaque cases, the SRS and spontaneous Raman results also matched by showing no such shift. We therefore agree that in this case SRS was not crucial for measuring the protein peak shift, but if available the SRS technique is of course much faster (see below*). Furthermore, SRS does not suffer from background fluorescence. On the other hand, spontaneous Raman offers full spectra and selectivity for carotenoids through pre-resonance enhancement. These arguments are now more clearly discussed on page 28 and for a proper comparison we would prefer to show both types of Raman results in the manuscript.

In addition, our reported SRS results, showing a β -shift in cored plaques, are a novel observation since previous publications on SRS of AD tissue were only done on transgenic mice tissue. This is the first time that a similar observation was found in human AD brain tissue.

*To map an area the size of 512 x 512 pixels the spontaneous Raman mapping acquisition would need around 218 hours. (That is also the main reason, why we are dependent on the auto-fluorescence image for localizing the plaque areas prior to Raman imaging to keep the Raman mapping times logistically acceptable.) In contrast, SRS is only able to scan one wavenumber at a time, but it only needs around 1 min to acquire data for the same image size. Even though we would need to scan around 20 different wavenumbers to cover the full Amide-I peak to highlight a β -shift, we would be more than two orders of magnitude faster to gain the same information.